# Tempo and Mode of Floristic Exchanges between Hainan Island and Mainland Asia: A Case Study of the *Persea* Group (Lauraceae)

**Xuejie Huo, Zhi Yang, Yinfeng Xie and Yong Yang ***

Center for Sustainable Forestry in Southern China, College of Biology and the Environment,
Nanjing Forestry University, 159 Longpan Road, Nanjing 210037, China
* Correspondence: yangyong@njfu.edu.cn

**Abstract:** The origin of island biodiversity has been a topic of interest in biogeography. Two main hypotheses were suggested to explain the floristic origin of Hainan Island: ancient vicariance vs. recent dispersal. The *Persea* group of Lauraceae was used to examine their origin on Hainan Island. A data matrix including five genera and 49 species was assembled, phylogenetic trees of the *Persea* group were reconstructed using both maximum likelihood and Bayesian inference methods, and a relaxed molecular clock in BEAST was used to estimate the divergence time of the *Persea* group. The results showed that the *Persea* group diverged from its sister clade in the Eocene, and that the endemic and common species of Hainan Island and mainland Asia originated relatively late during the Miocene–Pleistocene. Species of the *Persea* group on Hainan may have arrived from mainland Asia via dispersal or originated via dispersal–isolation–divergence during the Miocene and Pleistocene. The results favor the dispersal hypothesis for the origin of the flora of Hainan Island and negate the vicariance hypothesis.

**Keywords:** biogeography; flora; Hainan Island; Lauraceae; origin; *Persea* group

## 1. Introduction

Hainan Island is the largest tropical island in China. It is located south of Guangdong and Guangxi across the Qiongzhou Strait, east of Vietnam across the Beibu Gulf, and north of the Philippines, Malaysia, and Brunei [1]. The island is mountainous in the middle and slopes to the coast in all directions. It has a characteristic tropical monsoon climate (annual mean temperature: 22–26 °C), adequate light (annual mean hours: 2166 h), and abundant rain (annual mean precipitation: 2000–2400 mm) due to its geographic location [2]. The complex topography and favorable hydrothermal conditions have fundamentally shaped the rich floristic diversity [3].

The flora of Hainan has a strong tropical character [4]. Around 62.8% of the 196 families are tropical in nature. At the genus level, tropical elements account for 80.5%, while temperate elements are relatively low in proportion [3]. Hainan is rich in endemic species, 592 species endemic to China are on Hainan. Most of them are in the four mountainous areas of Wuzhi Shan, Jianfeng Ling, Bawang Ling, and Diaoluo Shan [4–6]. Around 2078 species are common to Asia in general.

Two hypotheses were proposed to explain the tempo and mode of the floristic origin of Hainan, although all studies agree that the flora of the island is continental in origin. Ancient vicariance is one hypothesis to explain the floristic origin of Hainan since the island was once connected to Vietnam and Guangxi, China, in the Eocene, then moved and rotated to the southeast to finally reach its present position [1,3]. The other hypothesis contends that the flora of Hainan was due to recent dispersal and frequent floristic exchanges with the Asian mainland [7–10]. These hypotheses have not been tested by phylogenetic studies based on molecular dating analyses.

Lauraceae are one of the early diverged lineages of angiosperms. Fossils of the family can be traced back to the mid-Cretaceous and have also been recorded in the Eocene flora of Hainan [11–13]. The family has fleshy fruits that are dispersed by birds, among other factors. It is mainly tropical and pantropical, and contains over 3000 species worldwide [14]. The family has the highest species diversity among characteristic tree families in Hainan [1,3] with 122 species and infraspecific taxa in 16 genera. Many species of Lauraceae are also dominant [15]. Within the family, the *Persea* group, with five genera, *Alseodaphne*, *Alseodaphnopsis*, *Dehaasia*, *Machilus*, and *Phoebe*, is well represented in the flora of Hainan. These genera contain species pairs between Hainan and continental Asia, thus providing an ideal opportunity to examine the hypotheses on the floristic origin of Hainan.

To test these hypotheses, we reconstructed a phylogeny of the *Persea* group based on extensive sampling of species on Hainan and mainland Asia and estimated the divergence time of the species. We expected the divergence time between Hainan and the adjoining Asian mainland to be earlier than the Eocene under the tectonic drift and vicariance hypothesis, but later than the Eocene under the recent dispersal hypothesis.

## 2. Materials and Methods

### 2.1. Taxon Sampling

We selected 49 species from five genera of the *Persea* group, with *Persea* excluded because it is not native to Hainan. Six species from three genera (*Lindera*, *Litsea*, *Neolitsea*) of the core Lauraceae were chosen as outgroups based on recent molecular studies of Lauraceae [16,17]. Two nuclear fragments, including the nuclear ribosomal internal transcribed spacer (nrITS) and the second intron of the *LEAFY* gene (*LEAFY* intron II), were selected for phylogenetic reconstruction [18]. GenBank accession numbers for nrITS and *LEAFY* intron II sequences are listed in Table 1.

**Table 1.** GenBank accession numbers for nrITS and *LEAFY* sequences.

| Taxon | nrITS | *LEAFY* |
|:---:|:---:|:---:|
| **Ingroups** | | |
| *Alseodaphne huanglianshanensis* H.W.Li & Y.M.Shui | HQ697182 | HQ697007 |
| *Alseodaphne semecarpifolia* Nees | HQ697184 | HQ697015 |
| *Alseodaphnopsis andersonii* (King ex Hook. f.) H.W. Li & J. Li | FM957793 | HQ697002 |
| *Alseodaphnopsis hainanensis* (Merr.) H.W. Li & J. Li | MG188587 | HQ697006 |
| *Alseodaphnopsis maguanensis* L.Li & J.Li | MN906900 | MN906896 |
| *Alseodaphnopsis petiolaris* (Meisn.) H.W.Li & J.Li | FM957796 | HQ697008 |
| *Alseodaphnopsis putaoensis* L.Li, Y.H.Tan & J.Li | MN906902 | MN906898 |
| *Alseodaphnopsis rugosa* (Merr. & Chun) H.W.Li & J.Li | MG188584 | HQ697011 |
| *Alseodaphnopsis sichourensis* (H.W.Li) H.W.Li & J.Li | MG188597 | MG188626 |
| *Alseodaphnopsis ximengensis* H.W.Li & J.Li | MG188591 | MG188599 |
| *Dehaasia hainanensis* Kosterm. | FJ719308 | HQ697025 |
| *Dehaasia incrassata* (Jack) Nees | HQ697186 | HQ697028 |
| *Machilus breviflora* (Benth.) Hemsl. | FJ755434 | HQ697041 |
| *Machilus decursinervis* Chun | AY934893 | HQ697044 |
| *Machilus duthiei* King ex Hook.f | FJ755425 | HQ697055 |
| *Machilus gamblei* King ex Hook.f | FJ755422 | HQ697037 |
| *Machilus gongshanensis* H.W.Li | FJ755416 | HQ697047 |
| *Machilus grijsii* Hance | FJ755420 | HQ697048 |
| *Machilus kwangtungensis* Y.C.Yang | FJ755424 | HQ697051 |
| *Machilus leptophylla* Hand.-Mazz. | FJ755430 | HQ697053 |
| *Machilus minutiflora* (H.W.Li) L.Li, J.Li & H.W.Li | HQ697208 | HQ697147 |
| *Machilus monticola* S.K.Lee | FJ755418 | HQ697056 |
| *Machilus nanmu* (Oliv) Hemsl. | FJ755409 | HQ697066 |
| *Machilus oculodracontis* Chun | HQ697188 | HQ697059 |
| *Machilus oreophila* Hance | FJ755423 | HQ697063 |
| *Machilus phoenicis* Dunn | FJ755413 | HQ697064 |
| *Machilus platycarpa* Chun | FJ755421 | HQ697067 |
| *Machilus pomifera* (Kosterm.) S.K.Lee | FJ755432 | HQ697069 |

**Table 1.** *Cont.*

| Taxon | nrITS | *LEAFY* |
|---|---|---|
| *Machilus robusta* W.W.Sm. | FJ755426 | HQ697071 |
| *Machilus salicina* Hance | FJ755428 | HQ697073 |
| *Machilus salicoides* S.K.Lee | FJ755433 | HQ697074 |
| *Machilus shweliensis* W.W.Sm. | FJ755414 | HQ697075 |
| *Machilus thunbergii* Siebold & Zucc. | HQ697190 | HQ697081 |
| *Machilus yunnanensis* Lecomte | FJ755415 | HQ697083 |
| *Phoebe angustifolia* Meisn. | HQ697201 | HQ697124 |
| *Phoebe tavoyana* Hook.f. | HQ697202 | HQ697130 |
| *Phoebe formosana* (Hayata) Hayata | HQ697205 | HQ697136 |
| *Phoebe hungmoensis* S.K.Lee | HQ697206 | HQ697137 |
| *Phoebe lanceolata* (Nees) Nees | FJ755410 | HQ697141 |
| *Phoebe macrocarpa* C.Y.Wu | FJ755408 | HQ697142 |
| *Phoebe megacalyx* H.W.Li | HQ697207 | HQ697144 |
| *Phoebe neurantha* (Hemsl) Gamble | HQ697209 | HQ697151 |
| *Phoebe puwenensis* W.C.Cheng | HQ697210 | HQ697152 |
| **Outgroups** | | |
| *Lindera erythrocarpa* Makino | HQ697215 | HQ697167 |
| *Lindera megaphylla* Hemsl. | HQ697216 | HQ697171 |
| *Litsea auriculata* S.S.Chien & W.C.Cheng | HQ697217 | HQ697174 |
| *Litsea verticillata* Hance | HQ697218 | HQ697175 |
| *Neolitsea cambodiana* Lecomte | HQ697219 | HQ697176 |
| *Neolitsea howii* C.K.Allen | HQ697220 | HQ697178 |

### 2.2. Phylogenetic Analyses

The nrITS and *LEAFY* intron II sequences were aligned with the program MAFFT (Version 7.471, Tokyo, Japan) [19] and edited manually using BioEdit (Version 7.0.9.0, Wooster, OH, USA) [20]. Ambiguously aligned fragments of two alignments were removed in batches using Gblocks (Version 0.91b, Barcelona, Spain) [21]. The sequences were concatenated and analyzed further. The best-fit partition model was chosen for the dataset (nrITS and *LEAFY* intron II) with ModelFinder [22] based on the Bayesian information criterion (BIC). Phylogenetic analyses were performed using the maximum likelihood (ML) and Bayesian inference (BI) methods. ML phylogenies were inferred using IQ-TREE (Version 1.6.8, Vienna, Austria) [23] under the edge-linked partition model for 5000 ultrafast bootstraps [24], and the Shimodaira–Hasegawa-like approximate likelihood ratio test [25]. BI phylogenies were inferred using MrBayes (Version 3.2.6, Stockholm, Sweden) [26], and the Markov chain Monte Carlo (MCMC) algorithm was run for 500,000 generations with a sampling frequency of every 500 generations. The initial 25% of sampled data were discarded as burn-in. Branch support of the BI tree was determined as Bayesian posterior probabilities (BPs).

### 2.3. Divergence Time Estimation

The combined dataset of the BI tree was used for molecular dating analyses with BEAST (Version 2.6.6, Auckland, New Zealand) [27]. We used BEAUti (Version 2.6.6, Auckland, New Zealand) [27] to import the dataset, set the substitution model as GTR, implemented the relaxed clock log-normal, and applied a Birth–Death Model. We ran the analysis for 40,000,000 Markov Chain Monte Carlo (MCMC) generations with a sampling frequency of every 4000 generations. TRACER (Version 1.7.2, Edinburgh, UK) [28] was used to calculate the log file's stationarity. After removing the first 10% of trees as burn-in, we generated a maximum clade credibility (MCC) tree in TreeAnnotator (Version 2.6.3, Edinburgh, UK) [28] and visualized it in FigTree (Version 1.4.2, Guangzhou, China) [29,30].

We used two macrofossils to calibrate divergence time estimates: *Alseodaphne changchangensis* JH Jin & JZ Li from the Eocene Changchang Formation of the Changchang Basin of Hainan [12] and *Machilus maomingensis* JH Jin & B Tang from the Eocene Youganwo Formation of the Maoming Basin of Guangdong, southern China [13]. We followed Li et al. [29]

in using *Alseodaphne changchangensis* to calibrate the crown node of the *Persea* group [A: age 37–49 million years ago (Ma)] and applied parameters including a log-normal prior distribution with an offset of 37 Ma, a mean of 1.8, and a standard deviation of 0.35. We used *Machilus maomingensis* to calibrate the stem age of *Machilus* (B: age 33.7–33.9 Ma) following Li et al. [29] and applied parameters including a uniform prior distribution with an offset of 0, a lower of 33.7, and an upper of 33.9 (Table 2).

**Table 2.** Fossil reference points used in this study.

| Node | Calibration Fossil | Minimum Age (MA) | Prior Distribution | Prior Parameters | 2.5/Median/97.5% Quantiles (Ma) |
|---|---|---|---|---|---|
| A: crown node of the *Persea* group | *Alseodaphne changchangensis* | 37–49 | log-normal | offset:37; M:1.8; SD:0.35 | 40/43/49 |
| B: steam node of the *Machilus* | *Machilus maomingensis* | 33.7–33.9 | uniform | offset:0; Lower:33.7; Upper:33.9 | 33.7/33.8/33.9 |

## 3. Results

### 3.1. Sequence Characters and Phylogenetic Analyses

#### 3.1.1. Sequence Characters

The numbers of variable sites and parsimony-informative (PI) sites of the nrITS dataset were 114 bp (23.1%) and 72 bp (14.5%), respectively (Table 3). The aligned length of the LEAFY intron II was 625 bp, with 24.2% and 16.7% variable and PI sites, respectively (Table 3). The aligned length of the combined nrITS and *LEAFY* intron II was 1088 bp, with 30.3% and 14.4% variable and PI sites, respectively (Table 3).

**Table 3.** Characteristics of separate and concatenated sequence datasets and the model selected for ML/BI analysis.

| Datasets | No. of Taxa | No. of Sites | No. of Variable/Parsimony-Informative Sites | ML Analysis | BI Analysis |
|---|---|---|---|---|---|
| nrITS | 49 | 496 | 114/72 | TPM2u+F+R3 | TPM2u+F+R3 |
| *LEAFY* | 49 | 625 | 231/96 | HKY+F+G4 | HKY+F+G4 |
| Combined | 49 | 1088 | 330/157 | Partitioned | Partitioned |

#### 3.1.2. Phylogenetic Analyses

The ML tree based on the nrITS and *LEAFY* intron II sequences (Figure 1) showed that the *Persea* group was monophyletic and divided into four clades with very high support (BS: 100; PP: 1). In the *Machilus* clade, *M. pomifera* and *M. monticola*, both endemic to Hainan, were grouped with *M. pomifera* and *M. salicoides* from southern China (BS: 100; PP: 1), while the phylogenetic position of *M. monticola* was not resolved. *Machilus grijsii*, distributed on Hainan and in southeastern China, grouped with *M. platycarpa* and *M. yunnanensis* from southwest China and the Indochina peninsula (BS: 98; PP: 1); *Machilus nanmu* was the earliest diverged species within *Machilus* (BS: 92, PP: 1). In the *Alseodaphne* and *Dehaasia* clade, *Alseodaphne* and *Dehaasia* were mixed together with moderate to high support (BS: 87, PP: 0.99). *Alseodaphnopsis* constituted a small clade, in which *Alseodaphnopsis hainanensis* from Hainan was sister to *Alseodaphnopsis putaoensis* from Southeast Asia (BS: 100, PP: 1). *Alseodaphnopsis rugosa* from Hainan was sister to *Alseodaphnopsis maguanensis* from Yunnan. In the *Phoebe* clade, *P. hungmoensis* from Hainan was clustered with species from Yunnan and adjacent countries, including *P. puwenensis*, *P. tavoyana*, *P. megacalyx*, and *P. macrocarpa*, with moderate to high support (BS: 89, PP: 0.99).

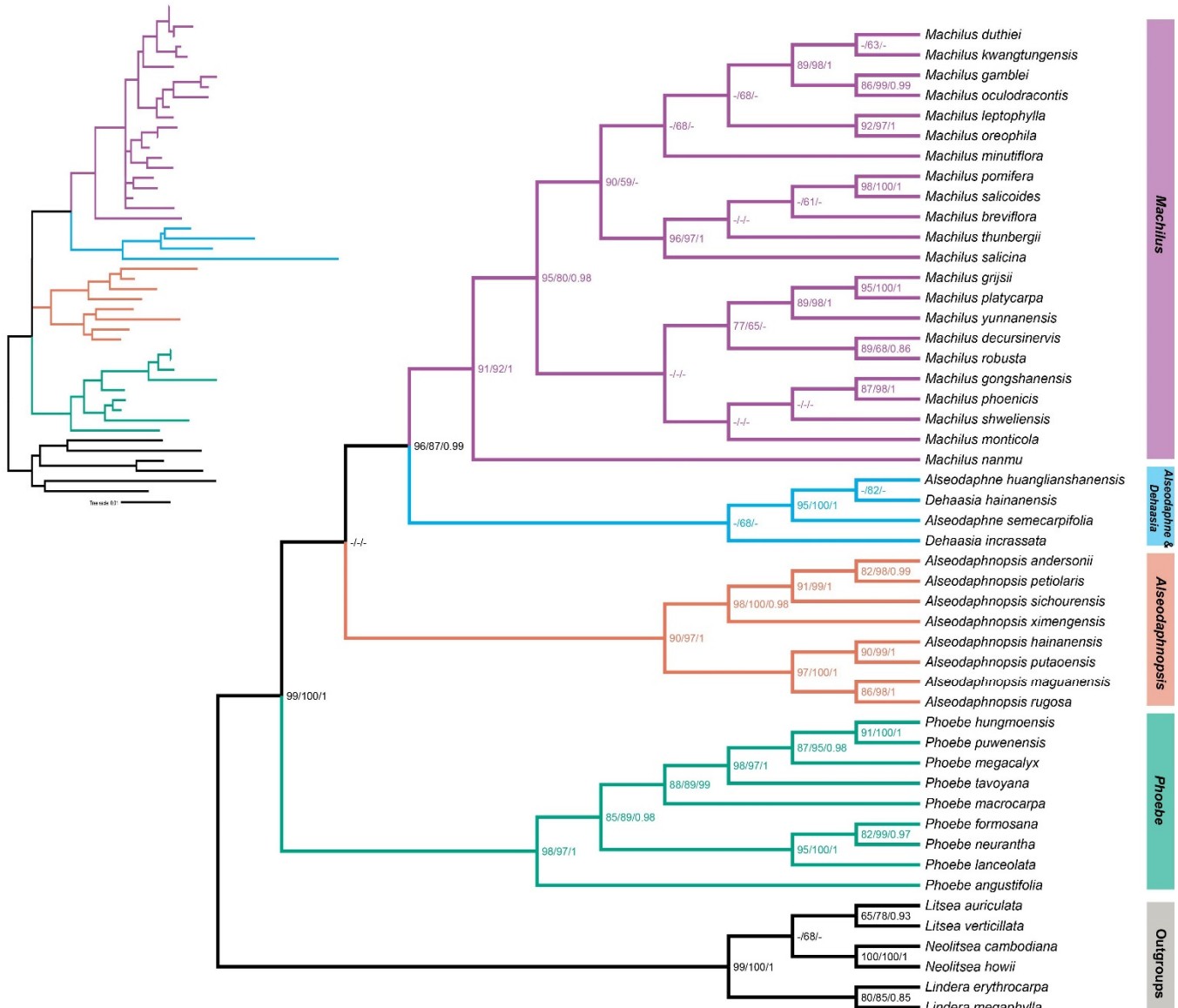

**Figure 1.** Maximum likelihood (ML) tree of the *Persea* group based on nrITS + *LEAFY* intron II. Numbers of nodes indicate support values of Shimodaira–Hasegawa-like approximate likelihood ratio test (SH-aLRT)/ML bootstrap support (BS)/Bayesian inference (BI) posterior probability (PP). "–" represents nodes with SH-aLRT/BS/PP support < 50%/0.8.

*3.2. Divergence Times*

The stem and crown ages of the *Persea* group were estimated to be 48.3 Ma (95% highest posterior density (HPD): 40.6–61.3) and 42.3 Ma (95% HPD: 39.6–46.3), respectively (Figure 2). *Alseodaphnopsis* was the earliest diverged lineage within the *Persea* group. The stem and crown ages were 42.7 Ma (95% HPD: 39.6–46.3) and 31.1 Ma (95% HPD: 22.6–40.1), respectively. The stem and crown ages of *Phoebe* were 40.2 Ma (95% HPD: 36.1–44.3), and 28.0 Ma (95% HPD: 18.7–37.3), respectively. The stem and crown ages of *Alseodaphne–Dehaasia* were estimated to be 33.8 Ma (95% HPD: 33.7–33.9) and 23.8 Ma (95% HPD: 17.4–29.9), respectively. The stem age of *Machilus* was the same as *Alseodaphne–Dehaasia*. The crown age was 25.8 Ma (95% HPD: 19.5–32.1).

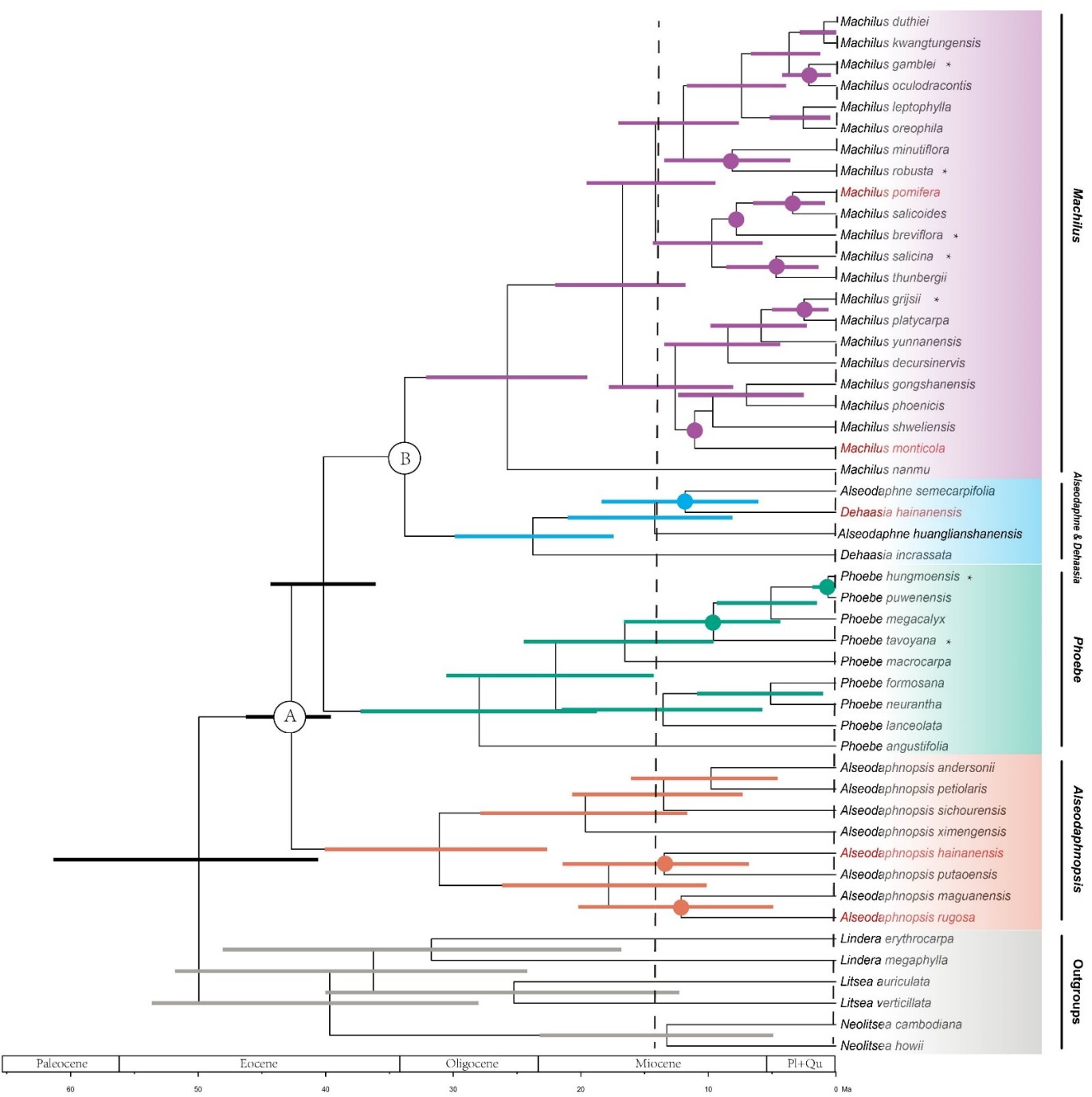

**Figure 2.** Time tree of the *Persea* group on Hainan based on nrITS and *LEAFY* intron II inferring from BEAST analysis. Letters in empty circles indicate fossil calibrations; species with asterisk (*) means occurring on Hainan; Hainan Island endemic species are highlighted in red; black bars show 95% confidence interval.

Endemism on Hainan arose around 13.5 Ma. *Alseodaphnopsis hainanensis*, the earliest diverged endemic species on Hainan, split from *Alseodaphnopsis putaoensis* around 13.5 Ma (95% HPD: 6.8–21.4). *Alseodaphnopsis rugosa* (Hainan endemic) split from its sister species around 12.1 Ma (95% HPD: 4.9–20.2). *Alseodaphnopsis putaoensis* and *Alseodaphnopsis maguanensis* are in Burma and Yunnan, China, respectively. *Dehaasia hainanensis*, which is endemic to Hainan, diverged from *Alseodaphne semecarpifolia* around 11.8 Ma (95% HPD: 6.1–18.4). *Machilus monticola* diverged from its sister species in Yunnan around 11.1 Ma (95% HPD: 8.1–17.9). The divergence time for *M. pomifera* was 3.4 Ma (95% HPD: 0.9–6.5); *M. pomifera* is sister to a

small clade, including *M. salicoides*, *M. breviflora*, *M. salicina*, and *M. thunbergii* of mainland China and Southeast Asia.

## 4. Discussion

Hainan was thought to be separated from the mainland (Beibu Gulf) due to Eocene plate tectonic movement, which fundamentally impacted the origin of the flora of Hainan [3,31]. Based on paleomagnetism and volcanism evidence, Zhu [31] proposed that the flora of Hainan originated via ancient vicariance in the Eocene. Our dating results suggest that species of the *Persea* group on Hainan originated in the Miocene and later, showing a dispersal–isolation–divergence pattern [32]. The split between mboxemphAlseodaphnopsis hainanensis and its sister species *Alseodaphnopsis putaoensis* represents the earliest divergence, which was estimated to have occurred around 13.5 Ma. *Machilus pomifera* (endemic to Hainan) diverged from its sister, *M. salicoides*, around 3.4 Ma. That divergence was the latest event. However, the estimated divergence time for the endemic species of Hainan from sister species was later than the geographic time of separation of Hainan Island from the mainland. We therefore hypothesize that species of the *Persea* group on Hainan may have been derived from mainland Asia via multiple dispersal events followed by isolation and speciation. Divergence of the species pairs between Hainan and the mainland occurred at different times in the Neogene. All of these divergence events between species pairs on Hainan and the Asian mainland occurred in the Miocene or later, which negates the ancient vicariance hypothesis but supports the dispersal hypothesis on the origin of the Hainan flora. Additionally, it appears that endemism on Hainan originated via a dispersal–isolation–divergence pattern.

Our findings are corroborated by a number of recent studies on the divergence time of Hainan endemic species in other families, including Dipterocarpaceae, Magnoliaceae, Podocarpaceae, and Theaceae. The Dipterocarpaceae are thought to have migrated from India to SE China via SE Asia. The species endemic to Hainan (*Hopea hainanensis*) diverged around 23.0 Ma [33,34]. Dong et al. found that *Michelia shiluensis* (Magnoliaceae), an endemic species on Hainan, split from its sister species around 8 Ma [35]. Klaus et al. suggested that *Dacrydium pectinatum* and *Podocarpus annamiensis*, two endemic species on Hainan, diverged around 12.5 Ma and 10 Ma, respectively [36]. Yu et al. indicated that *Polyspora hainanensis* (Theaceae), endemic on Hainan, diverged from its sister species around 3.8 Ma [37]. These studies showed that the endemic species of Hainan evolved multiple times since the early Miocene, thereby supporting the hypothesis of recent dispersal rather than ancient vicariance as the origin of the Hainan flora.

Different mechanisms may have contributed to the origin of the endemic flora of Hainan. Geological events have provided opportunities for floristic exchange between Hainan and the Asian mainland. Periodic land bridges during the Pleistocene connected Hainan and southern China, allowing for frequent exchanges [38,39]. *Cycas taiwaniana* of Hainan, Guangdong, and Fujian, diverged quite recently [40,41] and may have migrated between the mainland and Hainan when a land bridge was available. Plants with fleshy fruits, e.g., Lauraceae, may have entered Hainan via bird dispersal. A large number of avian fossils have been recorded from Miocene strata in Zhaotong, Yunnan Province [42]. Finally, ocean currents may have provided an additional mechanism for floristic exchange, e.g., *Cocos nucifera*, between Hainan and mainland Asia, including Southeast Asia and southern China.

## 5. Conclusions

The tempo and mode of floristic exchanges between Hainan and mainland Asia are complicated. We conducted phylogeny and molecular dating of the *Persea* group of Lauraceae to test the competing hypotheses of the origin of the flora of Hainan and concluded that the endemic species originated via multiple recent dispersal events, but not due to ancient vicariance. However, it should be acknowledged that we provided only a study of plants with easily dispersable fleshy fruits. To better understand the origin

of the flora of Hainan, further phylogenetic/phylogenomic and molecular dating and biogeographic studies are encouraged.

**Author Contributions:** Y.Y. conceived the idea. X.H. and Y.Y. designed the research. X.H. and Z.Y. collected data and performed analyses. X.H. drafted the manuscript, X.H., Z.Y., Y.X. and Y.Y. revised and finalized the manuscript. All authors have read and agreed to the published version of the manuscript.

**Funding:** This work was supported by the National Natural Science Foundation, China (31970205) and the *Metasequoia* funding of the Nanjing Forestry University.

**Data Availability Statement:** Not applicable.

**Acknowledgments:** We thank David E. Boufford of the Harvard University Herbaria for his kind help with English.

**Conflicts of Interest:** The authors declare no conflict of interest.

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
