# Peer review of "Tempo and Mode of Floristic Exchanges between Hainan Island and Mainland Asia: A Case Study of the Persea Group (Lauraceae)"

_forests, doi:10.3390/f13101722_

Round 1

Reviewer 1 Report

Very interesting article.  However I will suggest only some things to take into account .

The article is of great importance for studies of paleogeography, paleobotany, paleoecology and paleophylogenesis. The introduction is very complete and introduces the topic perfectly. The methods used are appropriate and the explanation of the results, despite being somewhat dense, is complete and perfect in structure and content. The discussion includes the essential aspects and it is appreciated that the work is short.

Suggestions for improvement:

1.- Explain very briefly, one or two sentences, what is meant by: “via dispersal–isolation–divergence” and add some bibliographical reference. especially not to confuse i with “Isolation with distance” (Wright, 1978) when the abstract is read.

2.- Could citations be included from scientific papers on fossil remains of birds in southern China in the Miocene? ... (paragraphs lines 204-213)

3.- Add one or two sentences about what is the dominant type of forest on the island and what are its characteristic species, are the species studied characteristic species of the island's forests? this is important for readers not familiar with this type of tropical vegetation

4.- Figures 1 and 2 do not have enough resolution to read the text. this is very important, the text they contain must be readable.

5.- Include in line 54 that the Lauraceae family has fleshy fruits and that it is dispersed, among other vectors, by birds.

6.- Line 68 after “Machilus decursinervis” add reference

7.- Line 185-186. “Via multiple dispersal events ” this sentence is a bit imprecise... try to be precise.

Author Response

Very interesting article.  However I will suggest only some things to take into account.

The article is of great importance for studies of paleogeography, paleobotany, paleoecology and paleophylogenesis. The introduction is very complete and introduces the topic perfectly. The methods used are appropriate and the explanation of the results, despite being somewhat dense, is complete and perfect in structure and content. The discussion includes the essential aspects and it is appreciated that the work is short.

REPLY. Thanks.

Suggestions for improvement:

1.- Explain very briefly, one or two sentences, what is meant by: “via dispersal–isolation–divergence” and add some bibliographical reference. especially not to confuse i with “Isolation with distance” (Wright, 1978) when the abstract is read.

REPLY. We used the term “dispersal-isolation-divergence” to address the sequential biological process to form the disjunct distribution of species pairs between the Hainan Island and the mainland Asia. Species originated in the mainland, and then spread to the Hainan Island by dispersal events (e.g. birds etc); then divergence occurred to form the species pairs between the Hainan Island and the mainland Asia because of isolation when immigration stops. This process is contrasting to the vicariance mode which caused the divergence of populations on the Hainan Island and the mainland Asia because of tectonic drift. We added one literature for this question (line 182). Thanks.

2.- Could citations be included from scientific papers on fossil remains of birds in southern China in the Miocene? ... (paragraphs lines 204-213)

REPLY. OK, we added one sentence in the Discussion on the fossil birds (lines 217-218), and one literature in the Reference (322-324). Thanks.

3.- Add one or two sentences about what is the dominant type of forest on the island and what are its characteristic species, are the species studied characteristic species of the island's forests? this is important for readers not familiar with this type of tropical vegetation

REPLY. OK, there is one sentence addressing this in our manuscript (line 59). Thanks.

4.- Figures 1 and 2 do not have enough resolution to read the text. this is very important, the text they contain must be readable.

REPLY. OK, we uploaded pdf files for the two figures. Thanks.

5.- Include in line 54 that the Lauraceae family has fleshy fruits and that it is dispersed, among other vectors, by birds.

REPLY. OK, done (lines 54-55). Thanks.

6.- Line 68 after “Machilus decursinervis” add reference

REPLY. This species name is only in Table 1, we are confused about this suggestion. Thanks.

7.- Line 185-186. “Via multiple dispersal events ” this sentence is a bit imprecise... try to be precise.

REPLY. OK, we added “via multiple dispersal events, divergence of the species pairs between the Hainan Island and the mainland Asia occurred at different times in the Neogene (lines 189-191) ”. Thanks.

Reviewer 2 Report

Manuscript titled Tempo and mode of floristic exchanges between Hainan Island and mainland Asia: a case study of the Persea group (Lauraceae) concerns an important problem related to the biodiversity of tree species on an island Hainan. Various hypotheses about the flora origin of Hainan Island are discussed.
Organization into sections was well done and paper structure is clear and arranged according to journal style.
 Introduction is understandable, clear and comprehensive exposing the reader to the topic.
Purpose and objectives are scientifically appropriate.
In “Introduction” are used 15 authors. 
This is sufficiently.
The methods used are accurate and correctly described. So described they allow experiment to be reproduced. Appropriate statistics was used. The experimental results are thoroughly and logically interpreted in the discussion and presented with clarity.
 Author (-s) compared the results obtained with the experiments and results of previous studies on the relevant subject. The findings are particularly valuable having in a mind importance of the this studied. The proposed peer review manuscript is of interest and it deserves to be published in the Journal Forests.

Author Response

Reviewer#2 gave no suggestions on revision. Thanks.